# Anxiety Screening among the General Population of Latvia and Associated Factors

**DOI:** 10.3390/medicina58091163

**Published:** 2022-08-26

**Authors:** Vineta Viktorija Vinogradova, Anda Kivite-Urtane, Jelena Vrublevska, Elmars Rancans

**Affiliations:** 1Department of Psychiatry and Narcology, Riga Stradins University, Tvaika Street 2, LV-1005 Riga, Latvia; 2Institute of Public Health, Riga Stradins University, Kronvalda Boulevard 9, LV-1010 Riga, Latvia; 3Department of Public Health and Epidemiology, Riga Stradins University, Kronvalda Boulevard 9, LV-1010 Riga, Latvia

**Keywords:** anxiety, generalized anxiety disorder, point prevalence, associated factors, epidemiology, general population

## Abstract

*Background and Objectives*: The aim of this study was to determine the point prevalence of at least mild anxiety symptoms and symptoms of generalized anxiety disorder in the Latvian general population, and to analyze the associated factors. *Materials and Methods*: A computer-assisted face-to-face survey was conducted in 2019–2020 with a multistage stratified probability sample of the Latvian general adult population (*n* = 2687). Anxiety was assessed using the 7-item Generalized Anxiety Disorder (GAD-7) scale; a score of ≥5 was defined as indicating the presence of mild symptoms of anxiety, and a score of ≥10 as the cutoff for identifying cases of generalized anxiety disorder. The Patient Health Questionnaire 9 (PHQ-9) and MINI International Neuropsychiatric Interview (M.I.N.I.) modules were used for assessing comorbid conditions. Multinomial logistic regression was conducted. *Results*: The point prevalence of mild anxiety symptoms was 10.9%. The point prevalence of generalized anxiety disorder symptoms was 3.9%. Higher odds of mild anxiety symptoms were detected in respondents of a young age (vs. 65 y.o. and older, aOR 3.1, *p* < 0.001), unmarried respondents (vs. married/cohabiting, aOR 1.5, *p* = 0.02), those living in the capital city (aOR 1.6, *p* = 0.008) or rural areas (aOR 1.5, *p* = 0.03) (vs. other towns), respondents with poor self-rated health (vs. good, aOR 2.6, *p* < 0.001), and diagnosed alcohol use disorder (aOR 1.9, *p* < 0.001), suicidal behavior (aOR 2.4, *p* < 0.001), and symptoms of depression (aOR 6.4, *p* < 0.001) (vs. no such conditions). As for symptoms of generalized anxiety disorder, female sex (vs. males, aOR 2.5, *p* = 0.003), age below 44 years (vs. 65+, aOR 6.2, *p* = 0.002), average self-rated health (vs. good, aOR 2.6, *p* = 0.005), and poor self-rated health (vs. good, aOR 5.3, *p* < 0.001), together with comorbid suicidal behavior (aOR 6.1, *p* < 0.001) and symptoms of depression (aOR 43.4, *p* < 0.001) (vs. no such conditions), increased the odds. *Conclusions*: Young age, poor self-rated health, and comorbid symptoms of depression and suicidal behavior were significant factors associated with symptoms of both mild anxiety and generalized anxiety disorder. Being unmarried, living in the capital city or rural areas, and alcohol use disorder were associated with mild anxiety symptoms alone. Female sex was associated with generalized anxiety disorder symptoms alone.

## 1. Introduction

Neurotic or anxiety disorders are the most common group of mental disorders worldwide, with 301 million prevalent cases globally in 2019 [1]; according to the data of large population-based European studies, these disorders constitute 14% of all diagnosed mental disorders in Europe [2]. The estimated newly diagnosed cases of anxiety disorders increased from 31.1 million in 1990 to 45.8 million in 2019 [1]. Moreover, in 2020–2021, due to the coronavirus disease 2019 (further COVID-19), the prevalence of anxiety disorders increased to 25.6% with the total prevalence of 4802.4 cases per 100,000 population globally [3].

Anxiety disorders usually include panic disorder, agoraphobia, generalized anxiety disorder (GAD), social anxiety disorder, specific phobias, post-traumatic stress disorder (PTSD), and obsessive–compulsive disorder (OCD).

Anxiety disorders are characterized mainly by nonspecific symptoms, such as restlessness, fatigue, poor concentration, irritability, muscle tension, and excessive worrying, and can be easily overlooked and underdiagnosed. Available data show that, indeed, only a minority of patients with anxiety disorders receive appropriate medical help, when effective treatment options exist [4,5]. In particular, 27.6% of patients with diagnosed anxiety disorder receive any treatment, and only 9.8% receive adequate guideline-concordant mental health care [6].

According to existing data, untreated anxiety can significantly impact the quality of life, especially the domain of social functioning [7], as well as the occupational and physical domains [8]. Associated levels of disability and reduction in mental quality of life exceed levels seen in patients with chronic physical conditions, such as heart disease or diabetes [9]. Untreated anxiety disorders are also associated with greater health care resource utilization [10].

Underdiagnosis of anxiety disorders seems to be a concerning problem for Latvia as well, since the data from the National Health Service Register show that the most prevalent diagnosed mental disorders in Latvia are organic mental disorders, schizophrenia spectrum disorders, and intellectual disability, but not neurotic and affective mental disorders, which are the most prevalent worldwide [11]. This allows us to hypothesize that most of the cases of anxiety disorders remain undetected in Latvia, especially taking into account that such a pattern was proved with depression: The only Latvian research of depression prevalence found that the point prevalence of depression in the general population of Latvia is 6.7% [12], which means that at least 115,000 cases of depression should be registered, but at the same time, in 2021, only 10,737 patients were diagnosed with a depressive episode or recurrent depressive disorder and registered in the National Health Service Register [11]. The general population of Latvia has never been screened in face-to-face interviews using a stratified random sample for anxiety symptoms and GAD symptoms. The only existing data concern the prevalence of GAD in primary care [13], which cannot be extrapolated to the general population.

Timely detection and treatment of anxiety are important for increasing the quality of life at a population level, decreasing social dysfunction and occupational disability, and reducing direct and indirect costs of this disability. For these purposes, screening programs can be helpful, but for effective use of resources, risk groups of the population should be defined precisely, and the screening algorithm should be specific and targeted. Therefore, the aim of our study was to determine the point prevalence of clinically relevant anxiety symptoms and generalized anxiety disorder symptoms, and to analyze associated factors in a representative sample of the Latvian general adult population. 

## 2. Materials and Methods

This cross-sectional quantitative study was carried out in Latvia from November 2019 to March 2020 and gathered information on a representative sample of the Latvian adult population (18 y.o. and older).

### 2.1. Sampling Procedure

The study sample was selected using a multistage stratified random sampling method and included 2687 adult persons (0.2%) from the 1.56 million residents of Latvia, which formed the target population in 2019. Stratification variables included age, sex, region, and urbanization level. A stratified sample of households was randomly selected from the address register administered by the State Land Service (SLS). The number of starting addresses varied according to the level of urbanization of settlements: 120 starting addresses in the capital city Riga, 10 to 16 starting addresses in large Latvian cities, 2 to 5 starting addresses in smaller towns, and one address in small counties. We included addresses that had not been previously used as starting addresses in the surveys conducted by the fieldwork agency for at least one year. Taking each of these addresses as a starting point, an additional 7 to 8 households were selected for interviews using a “random route method”: every household in rural areas and every second household in urban and semiurban areas. If, in the selected households, the residents were available for the initial contact, the choice of the respondent was based on the following principles:(1)If there was just one person meeting the inclusion criteria (residents of Latvia who have reached the age of majority and older), living alone, or being the only person at home, he or she was invited to participate in the survey;(2)If there were two or more people who corresponded to the target group, the principle of “younger man” was applied, meaning that from the several corresponding respondents, males with the youngest age in the household were chosen. This provided the choice of a respondent according to a certain system, and not according to the personal discretion of the interviewer or the people living in the household.(3)If no one at home met the inclusion criteria, but potential respondents lived in a particular household, it was clarified when at least one of them would be at home and the household was revisited;(4)If no initial contact was made, each household was revisited up to three times.

### 2.2. Data Collection

The survey was conducted using computer-assisted personal interviews (CAPI) in Latvian or Russian, according to the respondents’ choice. The fieldwork was carried out by 56 professional, specially trained and instructed interviewers. Before the fieldwork, training sessions on the methodology and theoretical background of the study were organized, and each interviewer received practical training on the scales used and the MINI International Neuropsychiatric Interview (M.I.N.I.) instrument, in particular. The fieldwork agency “KANTAR” followed the European Society’s of Marketing Research Professionals (ESOMAR) International Code on Market and Social Research, and the best professional practices of the Latvian Association of Sociologists.

### 2.3. Measures

The structured questionnaire consisted of three parts:(1)Questions about sociodemographic characteristics (age, sex, ethnicity, marital status, education, income), alcohol use, and smoking were asked by the interviewers. In this part of the questionnaire, respondents also evaluated their subjective health status (further self-rated health) by answering the question “How do you evaluate your current health status?” with possible answers: “bad”, “rather bad”, “average”, “good”, and “rather good”.(2)A self-completion part, which included the 7-item Generalized Anxiety Disorder (GAD-7) and 9-item Patient Health Questionnaire (PHQ-9) self-evaluation tools.(3)The MINI International Neuropsychiatric Interview (M.I.N.I) was performed by the interviewers, assessing 16 common mental disorders. For the purposes of the current manuscript, the results of the suicidal behavior, substance use disorder, and alcohol use disorder (further AUD) modules were used.

The prevalence of clinically relevant anxiety symptoms and generalized anxiety disorder was assessed by the 7-item Generalized Anxiety Disorder scale, which is a self-assessment screening tool with proven reliability for the detection of most common anxiety disorders, and good factorial, criterion, and procedural validity [14]. The good internal consistency and convergent validity of the GAD-7 have been pointed out in studies among primary care patients [15], patients with heterogeneous psychiatric conditions [16], and psychiatric outpatients [17]. More importantly, in the context of our study, the GAD-7 has also been proven to be reliable and valid as a measure of anxiety in the general population [18]. In the GAD-7, respondents are supposed to evaluate the previous two weeks and rate the frequency of particular anxiety symptoms as “not at all” (0 points), “several days” (1 point), “more than half of all days” (2 points), or “nearly all days” (3 points). The scale score can range from 0 to 21, and cutoff scores commonly used are 5, 10, and 15 for mild, moderate, and severe anxiety symptoms, respectively [15]. In our study, for detecting the presence of at least mild clinically relevant anxiety symptoms, we used a cutoff score of 5, and the presence of generalized anxiety disorder symptoms was defined using a cutoff score of 10 points. Evidence shows that for detecting GAD cases, a cutoff score of 10 has the optimal balance between sensitivity (89%) and specificity (82%) [14]. The GAD-7 translations for both Latvians and Russians have been used previously in the study of primary care settings in Latvia [13], and a validation manuscript has been recently submitted for publication.

For assessment of possible comorbid depressive symptoms, the 9-item Patient Health Questionnaire (PHQ-9) was used. The PHQ-9 assesses the previous 2 weeks before the interview and includes 9 questions based on diagnostic criteria for major depression from the Diagnostic and Statistical Manual of Mental Disorders, Fourth Edition (DSM-IV) [19,20]. For detecting clinically relevant depressive symptoms, we used a cutoff score of 10, which, as evidence shows, maximizes the combined sensitivity and specificity [21]. The PHQ-9 translations into the Latvian and Russian languages have been validated previously in a population of patients in primary care settings in Latvia in the frame of the National Research Project BIOMEDICINE (2014–2017) [22].

Possible comorbid alcohol use disorder, substance use disorder, and suicidal behavior were assessed using the appropriate modules of the MINI International Neuropsychiatric Interview (M.I.N.I.), Version 7.0.2. The M.I.N.I. is a structured diagnostic tool developed in accordance with the DSM-IV and the 10th version of the International Classification of Diseases *(ICD-10)* [23,24]. The M.I.N.I. version used in the current study has been translated into the Latvian and Russian languages by the authorship holders.

### 2.4. Statistical Analysis

Statistical data analysis was performed using International Business Machines Corporation’s (IBM corporation) Statistical Package for the Social Sciences IBM SPSS Statistics version for Windows 26.0, released 2019, (IBM corp., Armonk, NY, USA). Data were weighted by sex, age group, urbanization, region, and nationality. The total and stratified point prevalence rates of mild anxiety symptoms and GAD symptoms were calculated. Multinomial logistic regression was used to calculate the odds ratios (ORs) for both the univariate and multivariate logistic analyses. For the purposes of logistic regression analysis, we used the following groups of dependent variables concerning the GAD-7 score: <5 points, no symptoms of anxiety (reference category); 5–9 points, mild symptoms of anxiety; ≥10 points, symptoms of generalized anxiety disorder. All results are reported as ORs with 95% confidence intervals (CI). Statistical significance was considered as *p* < 0.05.

## 3. Results

The final weighted sample included 2687 respondents (46.1% (*n* = 1238) males and 53.9% (*n* = 1449) females). The median age of the respondents was 49.0 years (SD 18.2). Other basic sociodemographic characteristics of the sample together with stratified prevalence rates of mild anxiety and generalized anxiety disorder symptoms are summarized in Table 1.

The point prevalence of mild anxiety symptoms according to the GAD-7 (5–9 points) in the general population of Latvia was 10.9% (95%CI 9.4–11.6). The point prevalence of generalized anxiety disorder symptoms according to the GAD-7 (≥10 points) in the general population of Latvia was 3.9% (95%CI 3.2–4.6). We found that the point prevalence of mild anxiety symptoms (score ≥5) did not significantly differ between sexes: 10.8% among females (95%CI 8.8–11.8) and 11.0% among males (95%CI 9.2–12.6), *p* = 0.85. However, as for the point prevalence of generalized anxiety disorder symptoms, it was statistically significantly higher among females (4.9% (95%CI 3.8–5.9)) than among males (2.7% (95%CI 2.0–3.8)), *p* = 0.004.

As shown in Table 2, according to the univariate analysis, the odds of having mild clinically relevant anxiety symptoms were significantly higher in the youngest age group (vs. 65 y.o. and older, OR 2.2 (95%CI 1.5–3.1), *p* < 0.001). Higher odds of anxiety were detected in unemployed respondents (vs. economically inactive, OR 1.8 (95% CI 1.2–2.7), *p* = 0.002). According to the crude analysis, being unmarried was also associated with higher odds of mild anxiety symptoms (vs. married or cohabiting respondents, OR 2.1 (95% CI 1.6–2.8), *p* < 0.001), as were living in the capital city of Latvia, Riga (vs. living in other Latvian cities, OR 1.8 (95%CI 1.3–2.5), *p* < 0.001), and living in rural areas of Latvia (vs. living in other Latvian cities, OR 1.6 (95%CI 1.1–2.1), *p* = 0.006). High odds of anxiety symptoms were detected in respondents who evaluated their health as “bad or rather bad” (vs. “good or rather good”, OR 2.4 (95%CI 1.6–3.4), *p* < 0.001). Detected alcohol use disorder, according to the M.I.N.I., appeared to be a predictor of clinically relevant anxiety symptoms (vs. no diagnosed disorder, OR 2.4 (95%CI 1.7–3.2), *p* < 0.001). Similarly, the status of active smoker increased the odds of having anxiety (vs. nonsmokers, OR 1.5 (95% CI 1.1–1.9), *p* = 0.003). Additionally, according to the univariate analysis, especially high odds of having anxiety were found in depressed respondents, whose total PHQ-9 score indicated the existence of at least mild depressive symptoms (≥10 points) (vs. those with a PHQ-9 score <10 points, OR 10.3 (95%CI 6.9–15.5), *p* < 0.001). Similarly, detected suicidal behavior (vs. no suicidal behavior, OR 3.7 (95%CI 2.7–5.1), *p* < 0.001) and substance misuse disorder (vs. no such detected disorder, OR 3.5 (95%CI 1.7–7.2), *p* = 0.001), according to the M.I.N.I., appeared to be predictors of clinically relevant mild anxiety symptoms.

After adjustment for all independent variables (Table 2), young age (vs. 65 y.o. and older, aOR 3.1 (95%CI 1.7–5.6), *p* < 0.001) maintained its significance as an associated factor of anxiety. Being unmarried (vs. married or cohabiting, aOR 1.5 (95%CI 1.1–2.1), *p* = 0.02), living in the capital city (vs. other Latvian towns, aOR 1.6 (95%CI 1.1–2.2), *p* = 0.008), and living in rural areas of Latvia (vs. other Latvian towns, aOR 1.5 (95%CI 1.0–2.0), *p* = 0.03) also remained significantly associated with anxiety. After adjustment, “bad/rather bad” self-rated health status (vs. good, aOR 2.6 (95%CI 1.6–4.2), *p* < 0.001) was also significantly associated with higher odds of having anxiety symptoms. Diagnosed alcohol use disorder (vs. no diagnosed disorder, aOR 1.9 (95%CI 1.3–2.7), *p* < 0.001) increased the odds of being anxious. After adjustment for all independent variables, being depressed (vs. no depressive symptoms, aOR 6.4 (95%CI 4.0–10.3), *p* < 0.001), according to the PHQ-9 scale, maintained a very pronounced statistically significant increase in odds. Similarly, suicidal behavior diagnosed by the M.I.N.I (vs. no suicidal behavior, aOR 2.4 (95%CI 1.7–3.4), *p* < 0.001) remained significantly associated with mild clinically relevant anxiety.

According to the univariate analysis of factors associated with the detected generalized anxiety disorder cases according to the GAD-7 (see Table 3), female sex (vs. males, OR 1.9 (95%CI 1.2–2.8), *p* = 0.03), unfinished primary education (vs. higher education, OR 3.5 (95%CI 1.4–8.6), *p* = 0.006), being unemployed (vs. employed, OR 1.9 (95% CI 1.0–3.4), *p* = 0.04), and living separately or being divorced or widowed (vs. married or cohabiting, OR 1.6 (95%CI 1.0–2.5), *p* = 0.04) increased the odds of generalized anxiety disorder symptoms. Poor self-rated health state also appeared to be a predictor of generalized anxiety disorder symptoms: odds were higher in respondents who evaluated their health as “average” (vs. good, OR 2.4 (95%CI 1.5–4.0), *p* < 0.001), and especially high in those who rated their health as “bad/rather bad” (vs. good, OR 9.0 (95%CI 5.3–15.4), *p* <0.001). Detected alcohol use disorder (vs. no alcohol use disorder, OR 2.0 (95%CI 1.2–3.3), *p* = 0.005), according to M.I.N.I., and active smoking (vs. not smoking, OR 2.0 (95%CI 1.4–3.0), *p* < 0.001) increased the odds of having generalized anxiety disorder symptoms. As for depressive symptoms, an especially prominent increase in odds (vs. no depressive symptoms, OR 83.9 (95%CI 51.4–136.9), *p* < 0.001) of generalized anxiety disorder symptoms was observed. Additionally, in the univariate analysis, diagnoses of suicidal behavior (vs. no suicidal behavior, OR 14.7 (95%CI 9.7–22.2), *p* < 0.001) and substance use disorder (vs. no substance use disorder, OR 8.2 (95%CI 3.7–18.2), *p* < 0.001), according to the M.I.N.I., were associated with higher odds of generalized anxiety disorder symptoms

After adjustment for all independent variables (see Table 3), the odds of having generalized anxiety disorder symptoms were still statistically significantly higher in females (vs. males, aOR 2.5 (95%CI 1.4–4.6), *p* = 0.003) and respondents in the age group of 18–44 y.o. (vs. 65 y.o. and older, aOR 6.2 (95%CI 2.0–19.7), *p* = 0.002). Self-rated health state also maintained a significant association with generalized anxiety disorder symptoms: higher odds were found in respondents who rated their health as “average” (vs. good, aOR 2.6 (95%CI 1.3–4.9) *p* = 0.005) or “bad/rather bad” (vs. good aOR 5.3 (95%CI 2.2–12.5), *p* < 0.001). As for comorbid conditions, being depressed, according to the PHQ-9, after adjustment still maintained a significant association and especially high odds of generalized anxiety disorder symptoms (vs. no depressive symptoms, aOR 43.4 (95%CI 24.2–77.9) *p* < 0.001), as did suicidal behavior (vs. no suicidality, aOR 6.1 (95%CI 3.5–10.5), *p* < 0.001), but with a less dramatic increase in odds.

## 4. Discussion

There is a lack of data about the prevalence of anxiety among the general population of Latvia, which has never previously been screened for anxiety symptoms or generalized anxiety disorder. Conducted in a large, representative population-based sample, our research provides information about the point prevalence of clinically relevant anxiety and generalized anxiety disorder in Latvia, together with a comprehensive analysis of the associated sociodemographic factors and comorbidities before the COVID-19 pandemic. The only existing Latvian data about the prevalence of generalized anxiety disorder among patients of primary care settings in 2014 were published by members of our research group [25]. Their detected point prevalence of general anxiety disorder according to GAD-7 symptoms was 10.1%, but, due to the proved higher prevalence of GAD in primary care patients [26], this finding cannot be extrapolated to the general population of Latvia. Another Latvian study, published this year, assessed anxiety among the general population during the COVID-19 State of Emergency via an Internet survey, and according to the results, 15.2% of the participants were classified as having anxiety, but since anxiety was assessed during the extraordinary state of the pandemic, this finding also cannot be extrapolated to the general population in normal circumstances [27]. Moreover, a different methodological approach (online questionnaire) and diagnostic instrument (State–Trait Anxiety Inventory) were used, and the obtained data of both studies cannot be compared accurately. Our detected point prevalence of generalized anxiety disorder symptoms (3.9%) is in line with the published data of median prevalence rates of subthreshold GAD diagnoses in the general population (4.4%), according to a systematic review of epidemiological studies from Europe and North America [28]. There are not many studies assessing concepts such as the general prevalence of at least mild or mild symptoms of anxiety in the general population, since researchers usually focus on specific diagnostic categories, such as panic disorder or phobias. We determined that the point prevalence of mild anxiety symptoms is 10.8% (score  5−9  according to the GAD-7). We found just one epidemiological study with a similar methodology, but the researchers used a cutoff of 8 points and above, and their detected prevalence of anxiety in the general population was 8.2% [29]. The detected prevalence rate of mild anxiety symptoms is useful and novel, providing an overall comprehension of how large the proportion of the population suffering from mild anxiety is and what the associated factors are. Thus, screening strategies can be formulated, which could reveal anxiety symptoms maximally early, before the development of serious clinical conditions.

In our study, we have proven that generalized anxiety disorder symptoms are significantly more prevalent among women; this pattern has also been observed in other epidemiological studies in the general population [30]. Moreover, in primary care research in the UK, the prevalence of anxiety was almost twice as high in females as in males [31]. There can be several possible explanations for such a constant pattern of association. First of all, the neuroticism trait (an enduring tendency to worry and feel anxious) is one of the developmental and personality factors that are involved in the pathogenesis of anxiety, and according to the previously published research, it is more closely linked to anxiety in women than in men [32]. On the other hand, a recent genetic study found that genetic correlations between generalized anxiety disorder and neuroticism were high, but irrespective of gender [33]. Further investigation is needed. Some researchers suggest females are more prone than males to dysfunctional metacognitive beliefs about the uncontrollability of worries and maladaptive thought control strategies, such as punishment (punishing themselves for thinking the thought) and so called metaworry (“worrying about worrying”), which lead them to neurotic pathology [34]. Another convincing theory blames sex hormones—estradiol and progesterone, in particular—for promoting vulnerability to anxiety and the facilitation of its maintenance [35].

According to our results, younger age (18–44 y.o.) was significantly associated with higher odds of both mild anxiety symptoms and generalized anxiety disorder symptoms, in particular, and most of the recent studies support the finding of our research that anxiety is more prevalent in the youngest age group of adults [36,37]. Moreover, the latest research points out an interesting epidemiological dynamic: one study, conducted with the aim to assess the prevalence trends of anxiety among US adults from 2008 to 2018, suggested that anxiety increased from 2008 to 2018, and the most notable increase was among young adults, but not in those of age 50 and older [38]. The findings are partially consistent with data from Sweden reporting that young adults experienced an increase in self-reported poor sleep and bad general health from 2000 to 2016, but at the same time, overall stress and the level of loneliness were unchanged; in contrast, experiences of work-stress decreased [39]. This means that possibly factors other than overall stress or work-related stress are responsible for such tendencies in younger adults, such as the influence of social media. Research suggests that adolescents who spend more than 3 h per day using social media may be at heightened risk for mental health problems, particularly internalizing problems [40], and another study concluded that a greater amount of time spent on social media was associated with an increased risk of self-harm and depression [41]. It would be speculative to suggest that the younger generation is facing more stress-provoking world events than prior generations, but exposure to the information about such events is now more intrusive, unlimited in time, and pervasive through the media.

After adjusting for other factors, being unmarried maintained its significance as an associative factor of mild clinically relevant anxiety. In most studies, typically, being widowed, separated, or divorced is tied to anxiety spectrum disorder and generalized anxiety disorder, in particular [42], emphasizing the importance of previous partnership experience and subsequent loss of marital ties in developing such conditions. Our study shows the influence of being unmarried on the odds of developing mild anxiety without regard to the specific neurotic disorder. Our results can be partially explained by the impact of loneliness on the risk of developing anxiety [43].

Surprisingly, in addition to living in a big urban capital city (Riga) being tied to higher odds of anxiety—which has been previously prodigally discussed in the research and can be explained by the influence of overcrowded, polluted environments and high levels of stressors [44]—living in rural areas appeared to be associated with higher odds of anxiety as well. This induces the discussion that it is probably not only the urbanization grade itself that induces anxiety, but also some other factors. There are studies that suggest instead that neighborhood socioeconomic factors or lower social cohesion were associated with depressive and anxiety disorders [45]. This can be applied to the Latvian rural areas as well in regard to their lower socioeconomic status, reduced social security beneficiaries, and probably, in the era of virtual reality and the Internet, lower real and qualitative social interaction and support.

In the literature, one of the commonly mentioned risk factors of anxiety is low socioeconomic status [46], but we did not find in our study an association between income level and anxiety. After closer research of the literature, we concluded that socioeconomic status is usually measured as a combination of education, occupation, and income, and while variables such as education or social class exhibit a strong and consistent association with common mental disorders, others, such as income, show a more indefinite pattern of association. In a systemic review of the studies, the trends are the following: bivariate analyses show a relatively consistent positive association between low income and common mental disorders, but after adjustment for other independent variables, the association more vague [46]. There was a study that, due to the prospective design, was able to prove that household income by itself is not associated with increased risk of incident anxiety, but rather decrease in household income during a period of time is associated with an increased risk of mood, anxiety, or substance use disorders [47]. It makes us conclude that it is not the income level alone, but rather the changes in financial stability that are associated with common mental disorders and, in particular, anxiety. In our study, we did not investigate parameters of financial stress or changes in the income level over a period of time.

It is well known that poor self-rated health (SRH) predicts the symptoms of depression [48,49], and vice versa, poor physical health increases the risk of depression [50], but self-rated health has been less studied in regard to anxiety symptoms. SRH was studied in a population of older inmates in the US, and according to the results, worse SRH was associated with higher anxiety scores [51]. In our study, poor self-rated health increased the odds of mild anxiety symptoms 2.6 times and the odds of GAD symptoms 5.3 times. Owing to the cross-sectional design of our study, we cannot draw conclusions about the causality and direction of reciprocal associations between anxiety and poor health self-evaluation.

It has been reported that GAD is highly comorbid with substance misuse and other anxiety and mood disorders [52]. In the univariate analysis in our study, both alcohol use disorder and substance misuse were associated with higher odds of generalized anxiety disorder symptoms and mild anxiety, but after adjustment for all independent variables, only diagnosed AUD maintained its significance as an associated factor for anxiety symptoms—a finding consistent with previous surveys [52,53]. Prospective cohort studies are necessary to determine whether people with anxiety are more likely to become psychologically dependent on alcohol [54], whether there is a common genetic or biological pathway promoting the predisposition to both conditions, or whether there is an opposite causal direction and anxiety in AUD patients is a consequence of alcohol withdrawal [55].

The most pronounced increase in odds was related to the results of the PHQ-9 screening: being depressed increases the odds of anxiety 6.4 times and the odds of GAD symptoms even 43 times. Taking into account that most of the previous studies also concluded that anxiety is associated with developing a comorbid major depressive disorder [56,57], we should strictly take this into consideration in the development of screening strategies: All depressed patients should be screened for anxiety and vice versa. Moreover, the presence of a comorbid mood disorder significantly increases the risk of suicidal behavior in patients with anxiety disorders [58]. In published surveys, anxiety was associated with increased risk of suicide ideation and attempts [59]. In our study, diagnosed suicidality increased the odds of anxiety 2.4 times, but it increased the odds of GAD symptoms 6.1 times. This conclusion demonstrates that we must be aware of the timely detection of all three clusters of symptoms: depressive mood, anxiety, and suicidal behavior, because of such a close association between all of these conditions. Effective and evidence-based therapy for both anxiety and depression exists [4,5,60], which means that we can also reduce the risk of associated suicidality, which is especially important for Latvia, where the suicide mortality rate still remains one of the highest in the European Union (20.1/100,000 in 2019) [61].

The main strengths of our study include the large, nationally representative sample, the use of validated and internationally recognized measures for detecting anxiety, GAD symptoms, and comorbid conditions, specially trained interviewers, and face-to-face interviews. The fieldwork was conducted immediately before the COVID-19 pandemic and State of Emergency announcement in Latvia, which means that the obtained data can serve as a “point of reference” for studies comparing the prevalence of anxiety before and after the pandemic.

Several methodological limitations should be considered. Because of the cross-sectional design of this study, only conclusions on associations and not causation could be drawn. The use of the GAD-7 did not allow us to exclude anxiety due to substance misuse or organic/somatic causes. Voluntary recruitment may lead to a so-called nonresponse bias, where nonresponders might have different characteristics than survey respondents in regard to aspects other than basic sociodemographic data [62].

## 5. Conclusions

This is the first study that assesses the point prevalence of generalized anxiety disorder symptoms (3.9%) in the general adult population of Latvia, and it can be concluded that it is in line with studies of other European countries. The obtained data on the prevalence of mild clinically relevant anxiety symptoms, without a focus on specific diagnostic categories, are useful and novel, providing an overall comprehension of the proportion of the population suffering from mild anxiety (10.9%). Young age, poor self-rated health, comorbid depressive symptoms, and suicidal behavior were significant factors associated with both mild symptoms of anxiety and generalized anxiety disorder symptoms. Being unmarried, living in the capital city and rural areas, and alcohol use disorder were associated with mild anxiety symptoms alone. Female sex was associated with generalized anxiety disorder symptoms alone. Our results should be considered in developing effective and targeted screening strategies for anxiety disorders and planning targeted public health interventions that can help to detect clinically significant anxiety symptoms in a timely manner and reduce the no-treatment interval, which is associated with the economic burden.

## Figures and Tables

**Table 1 medicina-58-01163-t001:** Sociodemographic characteristics of the study sample, prevalence of anxiety symptoms in subgroups of the independent variables (*n* = 2687), weighted data.

	GAD-7 Score <5		GAD-7 Score 5–9		GAD-7 Score ≥10		Total *	
	*n*	%	*n*	%	*n*	%	*n*	%
Sex								
Female	1221	84.3	156	10.8	71	4.9	1448	53.9
Male	1068	86.3	136	11.0	34	2.7	1238	46.1
Age								
18–44 y.o.	927	81.7	159	14.0	48	4.2	1134	42.2
45–64 y.o.	792	86.5	88	9.6	36	3.9	916	34.1
≥65 y.o.	570	89.6	45	7.1	21	3.3	636	23.7
Education								
Higher	659	84.6	88	11.3	32	4.1	779	29.0
Secondary/ professional secondary	1342	86.7	153	9.9	52	3.4	1547	57.6
Primary	250	81.7	42	13.7	14	4.6	306	11.4
Unfinished primary	39	70.9	9	16.4	7	12.7	55	2.0
Ethnicity								
Latvian	1340	84.9	180	11.4	59	3.7	1579	58.8
Russian	753	86.2	84	9.6	37	4.2	874	32.5
Other	196	84.1	28	12.0	9	3.9	233	8.7
Employment								
Employed	1279	85.2	167	11.1	56	3.7	1502	55.9
Unemployed	234	78.8	45	15.2	18	6.1	297	11.0
Economically inactive (maternity leave, receiving disability benefits, etc.)	777	87.4	81	9.1	31	3.5	889	33.1
Marital status								
Married, cohabiting	1200	87.7	125	9.1	44	3.2	1369	51.0
Unmarried	419	78.5	93	17.4	22	4.1	534	19.9
Live separately, divorced, widowed	670	85.6	74	9.5	39	5.0	783	29.1
Place of residence								
Riga	745	82.6	120	13.3	37	4.1	902	33.6
Other city	841	88.7	74	7.8	33	3.5	948	35.3
Rural	704	84.1	98	11.7	35	4.2	837	31.1
Income								
No income	117	78.5	22	14.8	10	6.7	149	5.9
≤EUR 500/month	1124	85.0	142	10.7	57	4.3	1323	52.6
EUR 501–900/month	542	86.7	63	10.1	20	3.2	625	24.9
≥EUR 901/month	348	83.7	53	12.7	15	3.6	416	16.6
Self-rated health state								
Good or rather good	1231	88.4	136	9.8	26	1.9	1393	51.8
Average	871	85.2	107	10.5	44	4.3	1022	38.0
Bad or rather bad	188	69.1	49	18.0	35	12.9	272	10.2
Alcohol use disorder (M.I.N.I.)								
Yes	262	74.4	68	19.3	22	6.3	352	13.1
No	2028	86.9	224	9.6	83	3.6	2335	86.9
Smoking								
Active smoker	645	80.7	107	13.4	47	5.9	799	29.7
Nonsmoker	1645	87.1	185	9.8	58	3.1	1888	70.3
PHQ-9 score								
≥10 points	49	28.7	54	31.6	68	39.8	171	6.4
<10 points	2240	89.1	238	9.5	37	1.5	2515	93.6
Suicidal behavior (M.I.N.I.)								
Yes	164	57.5	65	22.8	56	19.6	285	10.6
No	2126	88.5	227	9.5	49	2.0	2402	89.4
Substance use disorder (M.I.N.I.)								
Yes	25	55.6	11	24.4	9	20.0	45	1.7
No	2265	85.7	281	10.6	96	3.6	2642	98.3

* The sum of responders may differ across variables due to missing values.

**Table 2 medicina-58-01163-t002:** Factors associated with mild anxiety (score 5–9 vs. no anxiety) in the univariate and multivariate analyses.

	Univariate Analysis			Multivariate Analysis		
Independent Variables	OR	95%CI	*p*	aOR *	95%CI	*p*
Sex						
Female	1.0	0.8–1.3	0.98	1.2	0.9–1.6	0.21
Male	1			1		
Age						
18–44 y.o.	**2.2**	**1.5–3.1**	**<0.001**	**3.1**	**1.7–5.6**	**<0.001**
45–64 y.o.	1.4	1.0–2.0	0.08	1.5	0.9–2.6	0.12
≥65 y.o.	1			1		
Education						
Higher	1			1		
Secondary/ professional secondary	0.9	0.6–1.1	0.29	0.8	0.6–1.1	0.14
Primary	1.3	0.8–1.9	0.25	0.9	0.6–1.5	0.76
Unfinished primary	1.8	0.9–3.9	0.11	1.9	0.8–4.5	0.16
Ethnicity						
Latvian	1			1		
Russian	0.8	0.6–1.1	0.19	0.9	0.6–1.2	0.33
Other	1.1	0.7–1.6	0.77	1.1	0.7–1.7	0.78
Employment						
Employed	1.2	0.9–1.6	0.12	1.1	0.7–1.7	0.78
Unemployed	**1.8**	**1.2–2.7**	**0.002**	1.4	0.8–2.4	0.20
Economically inactive	1			1		
Marital status						
Married, cohabiting	1			1		
Unmarried	**2.1**	**1.6–2.8**	**<0.001**	**1.5**	**1.1–2.1**	**0.02**
Live separately, divorced, widowed	1.1	0.8–1.4	0.69	1.2	0.8–1.7	0.34
Place of residence						
Riga	**1.8**	**1.3–2.5**	**<0.001**	**1.6**	**1.1–2.2**	**0.008**
Other city	1			1		
Rural	**1.6**	**1.1–2.1**	**0.006**	**1.5**	**1.0–2.0**	**0.03**
Income						
No income	1.2	0.7–2.1	0.42	0.9	0.4–1.7	0.67
≤EUR 500/month	0.8	0.6–1.2	0.28	1.0	0.6–1.6	0.94
EUR 501–900/month	0.8	0.5–1.1	0.17	0.8	0.5–1.2	0.21
≥EUR 901/month	1			1		
Self-rated health state						
Good or rather good	1			1		
Average	1.1	0.8–1.4	0.43	1.4	1.0–1.9	0.05
Bad or rather bad	**2.4**	**1.6–3.4**	**<0.001**	**2.6**	**1.6–4.2**	**<0.001**
Alcohol use disorder (M.I.N.I.)						
Yes	**2.4**	**1.7–3.2**	**<0.001**	**1.9**	**1.3–2.7**	**<0.001**
No	1			1		
Smoking						
Active smoker	**1.5**	**1.1–1.9**	**0.003**	1.2	0.9–1.6	0.20
Nonsmoker	1			1		
PHQ-9 score						
≥10 points	**10.3**	**6.9–15.5**	**<0.001**	**6.4**	**4.0–10.3**	**<0.001**
<10 points	1			1		
Suicidal behavior (M.I.N.I.)						
Yes	**3.7**	**2.7–5.1**	**<0.001**	**2.4**	**1.7–3.4**	**<0.001**
No	1			1		
Substance use disorder (M.I.N.I.)						
Yes	**3.5**	**1.7–7.2**	**0.001**	1.8	0.8–4.0	0.16
No	1					

* aOR: adjusted OR; adjustment was performed by including all variables mentioned in the table in one model; Statistically significant associations are bold.

**Table 3 medicina-58-01163-t003:** Factors associated with generalized anxiety disorder symptoms (score ≥10 vs. no anxiety) in the univariate and multivariate analyses.

Independent Variables	Univariate Analysis			Multivariate Analysis		
	OR	95%CI	*p*	aOR*	95%CI	*p*
Sex						
Female	**1.9**	**1.2–2.8**	**0.03**	**2.5**	**1.4–4.6**	**0.003**
Male	1			1		
Age						
18–44 y.o.	1.4	0.8–2.4	0.17	**6.2**	**2.0**–**19.7**	**0.002**
45–64 y.o.	1.3	0.7–2.2	0.39	1.8	0.7–4.8	0.23
≥65 y.o.	1			1		
Education						
Higher	1			1		
Secondary/ professional secondary	0.8	0.5–1.2	0.30	0.6	0.3–1.1	0.08
Primary	1.2	0.6–2.2	0.61	0.7	0.3–1.6	0.37
Unfinished primary	**3.5**	**1.4–8.6**	**0.006**	1.3	0.3–5.5	0.67
Ethnicity						
Latvian	1			1		
Russian	1.1	0.7–1.7	0.60	1.0	0.5–11.7	0.94
Other	1.0	0.5–2.1	0.95	0.9	0.3–2.1	0.74
Employment						
Employed	1.1	0.7–1.7	0.70	1.2	0.5–2.8	0.74
Unemployed	**1.9**	**1.0–3.4**	**0.04**	1.1	0.4–3.2	0.79
Economically inactive	1			1		
Marital status						
Married, cohabiting	1			1		
Unmarried	1.4	0.8–2.4	0.19	0.9	0.5–1.8	0.81
Live separately, divorced, widowed	**1.6**	**1.0–2.5**	**0.04**	1.4	0.7–2.7	0.36
Place of residence						
Riga	1.2	0.8–2.0	0.38	1.0	0.5–1.8	0.91
Other city	1			1		
Rural	1.3	0.8–2.0	0.35	1.1	0.6–2.1	0.74
Income						
No income	1.9	0.8–4.3	0.13	0.6	0.1–2.3	0.44
≤EUR 500/month	1.1	0.6–2.0	0.66	0.8	0.3–1.8	0.54
EUR 501–900/month	0.8	0.4–1.6	0.58	0.5	0.2–1.2	0.12
≥EUR 901/month	1			1		
Self-rated health state						
Good or rather good	1			1		
Average	**2.4**	**1.5–4.0**	**<0.001**	**2.6**	**1.3–4.9**	**0.005**
Bad or rather bad	**9.0**	**5.3–15.4**	**<0.001**	**5.3**	**2.2–12.5**	**<0.001**
Alcohol use disorder (M.I.N.I.)						
Yes	**2.0**	**1.2–3.3**	**0.005**	1.9	0.9–3.9	0.10
No	1			1		
Smoking						
Active smoker	**2.0**	**1.4–3.0**	**<0.001**	1.7	0.9–3.0	0.07
Nonsmoker	1			1		
PHQ-9 score						
≥10 points	**83.9**	**51.4–136.9**	**<0.001**	**43.4**	**24.2–77.9**	**<0.001**
<10 points	1			1		
Suicidal behavior (M.I.N.I.)						
Yes	**14.7**	**9.7–22.2**	**<0.001**	**6.1**	**3.5–10.5**	**<0.001**
No	1			1		
Substance use disorder (M.I.N.I.)						
Yes	**8.2**	**3.7–18.2**	**<0.001**	2.5	0.8–7.7	0.09
No	1					

* aOR: adjusted OR; adjustment was performed by including all variables mentioned in the table in one model; Statistically significant associations are bold.

## Data Availability

The data presented in this study are available on request from the corresponding author. The data are not publicly available because they contain sensitive information.

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
