# Peer review of "Anxiety Screening among the General Population of Latvia and Associated Factors"

_medicina, 2022, doi:10.3390/medicina58091163_

Round 1

Reviewer 1 Report

1. The latest statistical data can be presented in the introduction section.

2. Explaining 'the principle of “Younger Man” in the introduction section before usage will add value to the work

3. The previous partnership of males should also be one of the criteria similar to women's status as divorced, separated/Widowed

4. The job location of the spouse may also contribute to anxiety level status.

5. The conclusion section needs improvement with mention of points very specifically with facts and data obtained during the conduct of the study

6. The representation in the form of a Figure would be more fruitful and expressible. 

Reviewer 2 Report

Comments: The authors need to be commended to undertake this epidemiological cross sectional study of anxiety disorders. I understand that this is the first study to assess the point prevalence of generalized anxiety disorder symptoms in the general adult population of Latvia. Early detection and treatment of anxiety are important for improving the quality of life, limit social dysfunction and disability, thereby reducing the costs of disability. They used the validated GAD-7 in their study and assessed the point prevalence of clinically relevant anxiety symptoms and generalized anxiety disorder symptoms, and analyzed the associated factors in a representative sample of the Latvian general adult population. Like in other epidemiological studies, the authors found generalized anxiety disorder symptoms are significantly more prevalent among women. As is well known, the authors also found in their study that younger age (18–44 y.o.) was significantly associated with higher odds of both mild anxiety symptoms and generalized anxiety disorder symptoms. An important potential practical utility of their study results would be to design targeted screening strategies for anxiety disorders and planning targeted public health interventions that can help to detect clinically significant anxiety symptoms aimed at reducing the morbidity and economic costs of treatment.

Suggestions:

Page 3

Line 10: Under (1)What are the inclusion criteria

In Table 1:

Under Employment- what is the meaning of Economically inactive and how is it different from unemployed

Self-rated health state- Is it a validated scale

Page 6

2nd paragraph – All what is written is depicted in the table and this paragraph does not add any thing extra, therefore may consider deleting it

In discussion

What could be the explanation of some unusual findings as no correlation with income status ?

Reviewer 3 Report

In this study, Vinogradova et al aimed to determine the point prevalence of at least mild anxiety symptoms and symptoms of generalized anxiety disorder in the Latvian general population, and to analyze the associated factors. The manuscript is of good impact, well conceptualized, and clearly written.

Moreover:

  • Depression is not diagnosed on the basis of PHQ-9 alone. One cannot speak of depression, but rather depressiveness or symptoms of depression.

  • The abstract authors should be shorter.\

  • In the subsection 2.1 the authors should remove italics.

  • In the subsection 2.1 and Results section the authors should show percentage value.

  • The authors should describe the abbreviations of SPSS, KANTAR, ESOMAR,  AUD, COVID-19.
